# Hemp Flour Particle Size Affects the Quality and Nutritional Profile of the Enriched Functional Pasta

**DOI:** 10.3390/foods12040774

**Published:** 2023-02-10

**Authors:** Sonia Bonacci, Vita Di Stefano, Fabiola Sciacca, Carla Buzzanca, Nino Virzì, Sergio Argento, Maria Grazia Melilli

**Affiliations:** 1Department of Health Sciences, University Magna Græcia of Catanzaro, 88100 Catanzaro, Italy; 2Department of Biological, Chemical and Pharmaceutical Sciences and Technologies, University of Palermo, 90133 Palermo, Italy; 3CREA-Council for Agricultural Research and Economics-Research Centre for Cereal and Industrial Crops of Acireale, 00187 Roma, Italy; 4National Council of Research, Institute of BioEconomy (CNR-IBE), 95126 Catania, Italy

**Keywords:** pasta fortification, hemp flour, durum wheat cultivar, amino acids, fatty acids, mineral fortification

## Abstract

The rheological and chemical quality of pasta samples, which were obtained using the durum wheat semolina fortified with the hemp seed solid residue, after oil extraction, sieved at 530 μm (Hemp 1) or 236 μm (Hemp 2) at different percentages of substitution (5%, 7.5%, and 10%, were evaluated. The total polyphenolic content in hemp flour was quantified in the range of 6.38–6.35 mg GAE/g, and free radical scavenging was included in the range from 3.94–3.75 mmol TEAC/100 g in Hemp 1 and Hemp 2, respectively. The phenolic profiles determined by UHPLC-ESI/QTOF-MS showed that cannabisin C, hydroxycinnamic and protocatechuic acids were the most abundant phenolic compounds in both hemp flours. Among the amino acids, isoleucine, glutamine, tyrosine, proline, and lysine were the most abundant in raw materials and pasta samples. Although the hemp seeds were previously subjected to oil extraction, hemp flours retain about 8% of oil, and the fatty acids present in the largest amount were linoleic acid and α-linolenic acid. Characterization of the minerals showed that the concentration of macro and trace elements increased according to fortification percentage. Sensory evaluation and cooking quality indicated that the best performance in terms of process production and consumer acceptance was obtained using Hemp 2 at 7.5%. Hemp supplementation could be a potential option for producing high-quality, nutritionally rich, low-cost pasta with good color and functionality.

## 1. Introduction

In recent years, consumers’ eating habits have changed significantly. Food that, in the past, was intended to satisfy hunger and provide the necessary nutrients, today, combined with an active lifestyle, can help with harmonious physical and mental well-being.

Wheat is mainly used for the production of pasta, bread, and sweet and savory baked products. Pasta is one of the basic foods in the Mediterranean diet. Pasta traditionally made with durum wheat semolina can be prepared with “non-wheat flour” or by incorporating by-products from the agro-food industry, in variable percentages, which can increase its nutritional value [1,2]. Food by-products (grape, cereal bran, sunflower, artichoke, etc.) could represent interesting sources of bioactive compounds [3,4,5]. Fortification is the process by which nutrients with beneficial health effects are added to a food product in order to improve its nutritional quality and to increase its intake levels in the population. Food fortification (or enrichment) often negatively affects the quality of products, in terms of texture, color, cooking quality and sensory properties. Therefore, one of the main challenges in the food industry is to increase the healthiness of food without sacrificing sensory attributes [6].

Hemp (*Cannabis sativa* L.) is a plant used as textile fibers (from vegetative organs) native to the regions north and south of the Himalayas. Its use dates back to the Neolithic, and China is the country where it has been cultivated for the longest time. Its introduction in Europe probably dates back to the second millennium BC. Worldwide, it is cultivated mainly in Asia (China and India), Eastern Europe, and Russia. Today it is mainly used for textile, biocomposite, papermaking, construction, biofuel, and cosmetic purpose.

*Cannabis sativa* seeds can be used by the agri-food industry to produce flour, pastry, and oil, while the stem through to the canapulo (woody part of the stem) is in the green building sector. Its fiber (the external part of stem) will find new applications in the textile industry. Hemp inflorescences and roots, thanks to the extraction of bioactive molecules, will play an important role in the pharmaceutical and para-pharmaceutical industry [7].

Hemp seeds are mainly used as animal feed, but there is growing interest in their usage for human nutrition as a source of nutrients. They contain 25–35% of oil, 20–25% protein, 20–30% carbohydrates, and 10–15% insoluble fiber, vitamins, and minerals [8]. In particular, hemp seed oil is high in polyunsaturated fatty acids, with an ideal ratio (3:1) of linoleic acid (ω-6) and α-linolenic acid (ω-3) for human nutrition [9,10]. Merlino et al. [4] incorporated hemp seed flour (HSF) as a fortifying ingredient in the production of gnocchi, a typical Italian potato-based fresh pasta, from 5 to 20% HSF in substitution of soft wheat flour. Addition of HSF allowed for the enhancement of the nutritional value of gnocchi as a “source of fiber” in formulations with ≥ 10% of HSF. The fortified gnocchi had a high technological quality for cooking loss, cooking resistance, and textural properties, and average sensory quality; however, improving the HSF sensory quality for consumers’ satisfaction was necessary in terms of odor and bitter taste. Hemp seeds were used to enrich pasta (15%), and its effects on osteoarticular pain and bone formation markers in patients with osteoarthritis in post-arthroplasty rehabilitation were evaluated. The first results showed that hemp seed can improve pain symptoms in patients with osteoarthritis undergoing arthroplasty surgery and improves bone metabolism both in humans and in vitro [11]. Pasta samples fortified with 5–40% commercially available hemp flour or 2.5–10% of hemp cake obtained from hemp seed oil pressing were studied [12]. The addition of hemp seed raw materials led to an increase in the protein, total dietary fiber (TDF), ash, and fat contents in the pasta samples. Due to its lower granulation and higher nutritional value, hemp flour was found to be a better raw material for the fortification of pasta than hemp cake. 

In this study, the rheological and chemical qualities of pasta obtained by using the durum wheat cultivar “Ciclope”, fortified with different percentages of hemp flour (cv. Futura 75), were investigated. The influence of semolina replacement at different percentages (5%, 7.5%, and 10%) and using two hemp flours (with different particle sizes) was evaluated, highlighting the effect of the particle size of hemp flours on the sensory properties, cooking quality, mineral composition, and nutritional characteristics of cooked functional pasta.

## 2. Materials and Methods

### 2.1. Raw Material

The durum wheat cv Ciclope was chosen among the durum wheat varieties made up by CREA Research Centre for Cereal and Industrial Crops of Acireale (Catania–Sicily, Italy). Durum wheat grain sample was milled using an experimental laboratory mill (Bona Labormill 4RB, Monza, Italy) and fine particles were separated (250 µm), with an extraction rate of about 55–60%.

Hemp flours were obtained from cv Futura seeds provided by Mulino Crisafulli (Caltagirone, Catania, Italy). The hemp seeds were subjected to mechanical extraction of the oil; the residue was milled into powder using a blender model M20 (IKA, Staufen, Germany) and was kept in hermetic bottles at room temperature (20 ± 2 °C) and sieved at 530 µm (namely Hemp 1) and 236 µm (namely Hemp 2).

### 2.2. Pasta Making, Cooking Quality and Sensorial Analysis

A Ciclope semolina pasta sample (CTRL) and pasta fortified with variable percentages of hemp flour (Hemp 1 and Hemp 2 at 5, 7.5, and 10% of substitution) were prepared (Table 1).

Pasta samples were prepared using a Pastamatic ARIETE 1591 equipped with a mixer and an extruder (De Longhi Appliances s.r.l., Florence, Italy), mixing 500 g flour (durum wheat semolina + x% *w*/*w* of hemp flours) with distilled water for 10.5 min. in order to obtain a dough with 40% moisture. The dough was extruded into a mancherons shape (5 cm long) following the procedures described by Cardullo et al. [13]. The optimal cooking time (OCT), the cooking loss, and the amount of solid substance in the cooking water were evaluated according to the AACC-approved method 66–50 (2000). The swelling index of cooked pasta was determined according to the procedure described in previous papers [2,3,8,9,10].

In order to evaluate the sensory attributes, a panel of 8 trained tasters (4 men and 4 women, aged between 30 and 64 years) analyzed the cooked pasta in OCT. Panelists’ lists were developed on the basis of their sensory skills (ability to accurately determine and communicate sensory attributes such as the appearance, odor, flavor, and texture of a product). They judged bulkiness, firmness, adhesiveness, fibrous, color, odor, and taste. Based on the above-mentioned attributes, panelists were also asked to score the overall quality (OQS) of the product.

A 9-point scale was used: 1—very clear, 9—very dark in terms of color; 1—extremely unpleasant, −9—extremely pleasant in terms of bulkiness, firmness, adhesiveness, and fibrousness; 1—extremely unpleasant, 9—extremely pleasant in terms of odor, taste, and OQS [14,15,16].

### 2.3. Chemical Characterization

#### 2.3.1. Polyphenols Extraction

Phenolic compounds (PCs) can be classified as free, conjugated (to sugars and low molecular-weight compounds), and insoluble bound phenolics (BPs); these latter are covalently bound to the structural components of the cell wall [17,18]. BPs are not extractable in aqueous and/or organic solvents; therefore, preventive hydrolysis based on alkaline or acidic treatments is one of the most valuable strategies for targeting these compounds. The alkaline treatment can cleave the ester bonds linking the compounds to the cell wall, thus allowing for the release of PCs (mainly phenolic acids) from the insoluble residues. Free and bound phenolics were extracted using modified methods [19,20,21].

Eight grams of sample (Ciclope, Hemp 1 and Hemp 2 flours and ground, cooked pasta samples) were homogenized for 45 min in 40 mL 80% methanol solution using an ultrasonic bath. The samples were centrifuged at 5000× g for 15 min, and the supernatant was recovered. The pellet was re-extracted four times (repeating the protocol described above) and the supernatant was collected and evaporated using a rotary evaporator under vacuum at 45 °C. The residue was redissolved in 2 mL of methanol. This solution, containing free phenolic compounds (PCs), was filtered through a 0.22 µm nylon syringe filter into glass vials prior to HPLC-ESI/QTOF-MS analysis. In order to obtain the bound phenolics (BPs) extract, the residues separated after centrifugation were digested in 40 mL of NaOH 4 M for 1 h at room temperature and acidified using hydrochloric acid to pH 2. Subsequently, the acid solution was extracted with ethyl acetate (50 mL) four times and the organic fraction was evaporated in a rotary evaporator at a temperature of 45 °C. The residue was redissolved in 2 mL of methanol and solution filtered through a 0.22 µm nylon syringe filter prior HPLC-ESI/QTOF-MS analysis of bound phenolic (BPs) fraction. Both extractions for free (PCs) and bound phenolics (BPs) in samples were performed in triplicate.

#### 2.3.2. HPLC-ESI/QTOF-MS Analysis of Phenolic Compounds

The phenolic profile of hemp flour and pasta was investigated through an untargeted metabolomics-based approach using a HPLC-ESI/QTOF-MS method previously optimized [21]. The equipment consisted of an Alliance 2695 (Waters) HPLC system equipped with an autosampler, degasser, and column heater coupled with a Quadrupole Time-of-Flight (Waters Q-ToF Premier) mass spectrometer. The compounds were separated using a Phenomenex Luna C18 column (100 cm, 2 mm, 3 µm particle size). The phenolic compounds identified in different flours and pasta samples, were next quantified according to their class and sub-class, using calibration curves in a range of 2.5 μg mL^−1^–25 μg mL^−1^, built from pure reference standards (chlorogenic acid, catechin hydrate, rutin, caffeic acid, kaempferol, sinapic acid, and benzoic acid; Appendix A). When reference compounds were not available, the quantitation was based on structurally related substances. Specifically, rutine in negative mode was the reference compound for the determination of cannabisin B and cannaflavin C. Sinapic acid was the reference compound in negative mode for ferulic acid, chlorogenic acid was the reference compound for N-trans-caffeoyltyramine, and benzoic acid was the reference compound used for protocatechuic acid and vanillic acid semi-quantification.

#### 2.3.3. Total Phenolic Content (TPC)

The content of total phenolics (TPC) was determined using the Folin–Ciocalteau method [22]. A calibration curve was set with gallic acid ranging from 0.001 to 0.25 mg mL^−1^ methanol/water (80:20 *v*/*v*) (*y* = 10.955x + 0.1405, *R*^2^ = 0.992). The results were expressed as mg gallic acid equivalents per g (mg GAE g^−1^) of sample. In this method, 5 mL methanol/water (80:20 *v*/*v*) was added to 0.5 g of the flour samples (Hemp 1, Hemp 2, Ciclope flours, pasta samples obtained from semolina replacement with the two hemp flours at 5, 7.5, and 10%), then, the obtained mixture was filtered through a 0.45 μm PTFE syringe filter. Next, 125 μL of the solution was mixed with 625 μL of diluted (1:5) Folin–Ciocalteau reagent in water and 120 μL of 7% Na_2_CO_3_. The samples were left in the dark for 1 h at room temperature. The TPC was measured four times for each sample. 

#### 2.3.4. Fatty Acid Composition

After the basic hydrolysis of triglycerides, it was necessary to convert fatty acids into their methyl esters (FAMEs). Quali-quantitative determinations of FAMEs were conducted according to Melilli et al. [23] using a gas chromatography–mass spectrometry (GC/MS) ISQ™ 9000 Quadrupole GC-MS System (Thermo Fisher Scientific, Waltham, MA, USA). The identification of FAMEs was performed by comparing their retention times with those of reference standards (mixture FAME Mix, SUPELCO, which included 37 FAMEs). The results of the FAMEs were expressed as relative percentages (%).

#### 2.3.5. Amino Acids (AAs) Quantification by HPLC-FLD Method

Some procedures are needed for amino acid analysis, such as proteins hydrolysis. The modified procedure employing the acid hydrolysis of protein and the derivatization of the free amino acids using FMOC-Cl (9-fluorenylmethylchloroformate) was required prior to analysis with HPLC-FLD. Five hundred milligrams of the sample were added with 1 mL HCl 6 M in order to support the subsequent hydrolysis of proteins and were incubated in an oven at 110 °C for 24 h. After cooling to room temperature, the sample was diluted with 2 mL of deionized water and filtered with 0.45 μm PTFE syringe filters. The solution was subjected to pre-column derivatization by reaction of the sample with FMOC-Cl: 200 μL of 3 mM FMOC-Cl acetonic solution and 200 μL of borate buffer were added to 50 μL of the solution containing amino acids. The solution was heated at 70 °C for 10 min. Subsequently, 50 μL of a heptylamine solution (3 mL heptylamine, 15 mL ACN, and 175 mL HCl 0.1 M) was added to the solution and mixed for 3 min. Eighty microliters of the latter solution were taken, and 320 μL ACN and 600 μL hexane were added. A volume of 20 μL of this solution was injected into the HPLC-FLD instrument. Derivatized amino acids analyses were carried out using an HPLC Agilent 1100 Series chromatographic system equipped with a G1312A binary gradient pump and a fluorescence FLD detector and controlled by Chemstation software. For the chromatographic separation of derivatized amino, a Discovery HS C18 column was used (4.6 mm × 150 mm. 3.5 μm) (Supelco, Bellefonte, PA, USA), which fitted with guard column. The column operated at 40 °C, the flow rate was maintained at 1mL min^−1^. Mobile phases were 0.1% formic acid as eluent A and ACN as eluent B. The program of gradient elution was as follows: 0–10 min, 3% B; 3–17 min, linear increase to 10% B; 17–47 min, linear increase to 50% B; 47–57 min, linear increase to 100% B; 57–60 min, hold 100% B; 60–63 min, equilibration and return to the initial conditions. Each derivative eluted from the column was monitored by a fluorometric detector (FLD) set to an excitation wavelength of 254 nm and an emission wavelength of 630 nm. A comparison of the retention times of the standards for peak identification was carried out, and a fortification technique (spiking) was applied. Quantitative determination of the derivatized amino acids was performed using calibration curves. Standard solutions of the derivatized amino acids were prepared at five concentration levels in a range from 0.025 mM to 0.4 mM (Appendix A). The results were expressed in terms of grams of amino acids in 100 g of sample.

#### 2.3.6. Antiradical Properties of Raw Materials and Functional Pasta

The antiradical activity of samples (flours and fortified pasta) was measured using the DPPH assay.

One gram of each sample was extracted with 4 mL of methanol for 40 min in an ultrasonic bath. The supernatant was filtered using a 0.45 μm PTFE syringe filter. One hundred microliters of the filtrate were mixed with 3 mL DPPH (60 μM in methanol) and placed in the dark for 30 min. Absorbance at 515 nm was measured with a spectrophotometer (Varian Cary^®^ 50 UV-Vis spectrophotometer) using methanol as a blank. Antiradical scavenging activity was expressed as the percentage inhibition of the DPPH radical and was calculated using the following Equation:Scavenging% = (A0 − Ai/A0) × 100
where A0 is the absorbance of DPPH without the sample, and Ai is the absorbance of the sample and DPPH. The results were also reported as TEAC (Trolox equivalent antioxidant activity) and expressed in terms of mmol Trolox equivalents (TE)/100 g of sample. Trolox was utilized as the standard, and the calibration curve in a range between 5 and 400 μM was prepared using methanol as solvent (y = 0.0037x + 0.1655 and R2 = 0.987). All of the experiments were carried out in triplicate.

#### 2.3.7. Mineral Profile of Pasta

In order to assess the influence of cooking on the exchange of mineral contents, the elemental composition of the raw and cooked hemp pasta samples and of the different cooking waters was established. Elemental analysis of microelements (As, Be, Cd, Co, Hg, Li, Ni, Sb, Se, Sn, Sr, and V) was performed using an inductively coupled mass spectrometer ICP-MS iCAP RQ, (Thermo Fisher Scientific Inc., Bremen, Germany) operating with argon gas of spectral purity (99.999 sample solutions were pumped by a peristaltic pump from tubes arranged on a CETAC ASX-520 auto-sampler (Thermo Scientific, Omaha, NE, USA). Instrument sensitivity, resolution, and mass calibration were optimized daily with the tuning solution (iCAP Q/RQ Tune aqueous multielement standard solution (Thermo Scientific, Bremen, Germany) in order to maximize ion signals and minimize interference due to high oxide levels, optimizing torch position, ion lenses, gas output, resolution axis, and background. The optimal parameters are shown in Table 2.

The Al, B, Ba, Ca, Cu, Fe, Mg, Mn, Mo, Na, P, and Zn contents were determined using an inductively coupled plasma optical emission spectrometer (ICP-OES Analyzer, iCAP 7400, Thermo Fisher Scientific Inc., Waltham, MA, USA) equipped with a concentric nebulizer and a cyclonic spray chamber. The operating conditions are shown in Table 3.

Sample preparation was carried out using an Anton Paar Multiwave 5000 digestion system equipped with an XF100 rotor. In order to decontaminate PTFE vessels, a cleaning procedure was carried out by adding 4 mL of HNO_3_ and 4 mL of H_2_O to each vessel under the following conditions: 1100W for 15 min. After cleaning, vessels were rinsed with ultrapure water and dried [24]. Aliquots of 0.5 g of each pooled sample were weighted directly into the PTFE vessel of the microwave system. Digestion was performed by adding 8 mL of HNO_3_. The operating conditions used for the microwave digestion were 800 W over 15 min and then hold at this power for 30 min. After digestion, samples were quantitatively transferred to a graduated polypropylene test tube and diluted with ultrapure water to 50 mL and stored at 4 °C until analysis. Each sample’s digestion was performed in triplicate. The analytical batch consisted of a set of calibration standard samples and a minimum of three procedural blanks. Each solution was measured in triplicate, and analyses were carried out by a classical external calibration approach. For each element at least six calibration points were considered for calculation. The concentration range was selected based on the expected elemental values and sample dilution. The calibration ranges were: 0.005–100 μg L^−1^ for microelements; 0.002–1 mg L^−1^ for Ba, Cu, Mn, Mo, and Zn; 0.1–100 mg L^−1^ for Ca, Mg, Na, and K; and 0.01–10 mg L^−1^ for Al, B, Fe, and P. Stock solutions of calibration standards were properly diluted with 5% HNO_3_.

### 2.4. Data Analysis

Data were submitted to the Bartlett’s test for homogeneity of variance and then analyzed using two-way analysis of variance (ANOVA), based on a factorial combination of particle size (PS) × percentage of substitution (S) for the sensory characteristics of pasta. A separate ANOVA was conducted for each measurement and each main factor (particle size or percentage of substitution); in this case, means were statistically separated based on the Student–Newman–Keuls test. The CTRL was excluded when comparing pasta fortified with the two types of hemp flours. All other data, following Bartlett’s test for the homogeneity of variance, were analyzed using one-way ANOVA, and means were compared by *LSD* test when the *F*-test was significant, at least at the 0.05 probability (CoHort Software, CoStat version 6.451).

## 3. Results

### 3.1. Chemical Characterization of Raw Material

The chemical characterization of the raw material is shown in Table 4. The total phenolic contents (TPC) of Ciclope durum flour and Hemp 1 and Hemp 2 were examined. For Ciclope flour, total polyphenol content, determined using the Folin–Ciocalteau method, was 2.45 mg GAE/g. In the case of hemp flours, similar total polyphenol values were obtained (6.38 and 6.35 mg GAE/g, respectively, for Hemp 1 and Hemp 2). The DPPH method has been widely used in antiradical activity studies of plant extracts [25]. DPPH radical scavenging activity was also expressed as the % scavenging value. The results of the radical scavenging activity of sample flours showed that all extracts had the ability to scavenge DPPH radical with values of 29.7, 53.2, and 51.5 for Ciclope, Hemp 1, and Hemp 2, respectively. The results confirmed that Hemp 1 and Hemp 2 flours showed similar antiradical activity, which was higher than Ciclope durum wheat flour.

Although the hemp flours come from the shredding and sieving of the defatted seeds, a small percentage (about 8%) of oil remained in the solid matrix. The fatty acid profile was evaluated by GC-MS analysis and reported in Table 4. The main fatty acids identified in the lipid fractions of Ciclope flour were palmitic, oleic, and linoleic acids, with lower percentages of α linolenic acid. The lipidic profiles of hemp flours were shown in linoleic and α linolenic acid as major fatty acids, and oleic and palmitic acids as minor. Regarding the quantity of fatty acids, there was no differences in the lipid profile of Hemp 1 and Hemp 2 flours. The fatty acid present in largest amount was linoleic acid in 53% and α linolenic acid with 15.5%. Results were consistent with the analyses of other authors; Siano et al. [26] in agreement with our results, identified a similar fatty acid composition with linoleic acid as prevalent (56.42%), followed by linolenic (14.55%), oleic (12.79%), γ-linolenic (3.03%) and as saturated, palmitic (7.35%) and stearic acids (2.26%). Pojic work [27] highlighted high content of linoleic (54.09%–55.43%), linolenic (17.31–18.42%) and oleic (12.96–13.93%) acids, followed by palmitic (6.48–7.90%), stearic (3.18–3.86%) and γ-linolenic (2.61–2.76%).

There are no studies on the amino acid composition of Ciclope flour and hemp flour. Table 4 moreover summarizes the mean individual and total free amino acid (AAs) contents observed in the studied flours samples. According to the AAs profiles isoleucine, leucine, tyrosine and serine were among the amino acids with the highest content in Ciclope. The total AA content was 19.04 g/100 g in Ciclope. In Hemp 1 and Hemp 2 isoleucine, glutamine, tyrosine, proline and lysine were the most abundant. The particle size affected the amounts of the total AA content with 32.3 g/100 g (Hemp 1) and 34.4 g/100 g (Hemp 2). Particularly interesting was the amount of essential amino acids determined in the hemp flours which resulted in 15.9 g/100 g and 17.5 g/100 g in Hemp 1 and Hemp 2, respectively.

Hydroxycinnamic and protocatechuic acids (Table 5) represented the most abundant phenolic compounds quantified in hemp flour samples. The samples were subjected to alkaline treatment in order to determine the bound phenolic fraction (BPs). Results show a higher amount of bound hydroxycinnamic acid in Hemp 1 than Hemp 2 (1687.4 µg 100 g^−1^ and 1589.0 µg 100 g^−1^, respectively). The presence of cannaflavin C is highly relevant both in PCs and BPs form. Values ranging from 1384.1 to 3367.0 µg 100 g^−1^ for Hemp 1 and from 1139.2 to 2207.0 µg 100 g^−1^ for Hemp 2.

As far as phenol amides were concerned, N-trans-caffeoyltyramine was quantified uniquely in Hemp 2 as bound phenol (1817.1 µg 100 g^−1^). In Ciclope flour modest amounts of bound phenols have been determined, among these protocatechuic acid, *p*-hydroxycinnamic acid and vanillic acid (189.5, 329.6 and 75.2 µg 100 g^−1^, respectively). No free-form phenolic compounds were found.

### 3.2. Chemical Characterization of the Functional Pasta

Total phenolic contents and antiradical activity were determined on cooked Ciclope pasta and the fortified pasta samples as shown in Table 6. The data reveal TPC values in Hemp 1_10% and in Hemp 2_10% pasta samples (4.92 ± 0.31–4.21 ± 0.35 mg GAE/g, respectively), in agreement with the activity of radical scavenging (3.86 ± 0.07–3.14 ± 0.06 mmol TE/100 g, respectively), higher than Ciclope semolina pasta. 

Methyl ester fatty acids profile, as shown in Table 6, was also studied on cooked pasta samples. The main differences concern γ linolenic acid and α linolenic acid. In Hemp 1_10% and in Hemp 2_10% pasta samples reported, respectively, 5.26% and 5.71% of γ linolenic acid, and 15.89 of α linolenic acid.

Table 6 also highlights the amino acid content in the cooked pasta sample. Amino acids such as tyrosine, glutamine, proline and isoleucine are among the main in Hemp 1_10% and in Hemp 2_10% pasta. The concentration of essential amino acids in the two different pasta samples was interesting. In particular, there was a concentration of 4.30 g/100 g in Hemp 1_10% and 4.62 g/100 g in Hemp 2_10%. The concentration of total amino acids in the 10% fortified pasta was more than double that of the Ciclope durum wheat pasta.

In hemp-fortified pasta, phenols were detected and quantified (Table 7).

As expected, the alkaline treatment of the samples allowed for the cleavage of the ester bonds that bind the compounds to the cell wall, thus allowing for the release of PCs (mainly phenolic acids) from the insoluble residues.

In the pasta samples fortified, after the alkaline treatment was observed a greater presence of bound phenolics, particularly for the samples obtained with 10% substitution. Cannaflavin C, *p*-hydroxy benzoic acid, protocatechuic acid, hydroxycinnamic acid, and caffeic acid were predominantly found in Hemp 1 pasta in bound form. Conversely, *p*-hydroxycinnamic acid, caffeic acid, *p*-hydroxy benzoic acid, protocatechuic acid, and trans-N-caffeoyl-tyramine resulted in higher quantities in Hemp 2 pasta samples. As expected, the amount of free and bound phenols decrease the percentages of fortification.

The fortification at 7.5% especially with Hemp 1, showed good values of bound phenols. A similar phenolic profile was also reported by Pannico et al. [28] and Izzo et al. [29]. 

The quantification of minerals reported in Table 8 was carried out using external calibration curves. The data allow for the assessment of the contribution of hemp enrichment to the macro-element’s composition of pasta.

Most of the minerals’ concentrations increased according to the fortification percentage. In particular, the addition of hemp to pasta increased the content of iron, potassium, magnesium, and phosphorus. The iron concentration increased from 0.035 mg g^−1^ to 0.051 mg g^−1^, the potassium concentration increased from 1.953 mg g^−1^ to 2.020 mg g^−1^, the magnesium concentration increased from 0.873 mg g^−1^ to 1.191 mg g^−1^, and the phosphorus concentration increased from 2.502 mg g^−1^ to 3.086 mg g^−1^ in Hemp 1_5% pasta and Hemp 1_10% pasta, respectively. As for trace elements, the copper concentration increased from 7.903 ug g^−1^ to 9.669 ug g^−1^ in Hemp 1_5% pasta and Hemp 1_10% pasta, respectively. The same results were recorded in pasta fortified with Hemp 2. The mineral element content in hemp seeds was nutritionally interesting, as reported by Alonso et al., 2022 [30]. Phosphorus, potassium, magnesium, calcium, iron, zinc, manganese, and copper are essential dietary elements for mammals and are involved in many physiological processes [31]. In Figure 1 the percentage of variation is reported for the most important minerals in the hemp-enriched pasta vs the CTRL. Pasta enriched with Hemp_2 yielded the best results for all of the considered minerals except Fe.

### 3.3. Pasta Quality

Semolina particle size is a key quality factor in pasta making. Semolina used for pasta processing typically ranges in particle size from 550 to 150 μm [31]. The semolina used in this study had fine particles (<250 µm), similar to Hemp 2 flours. The addition of hemp flour significantly affected the sensory attributes of cooked pasta (Table 9). Substantial differences were recorded between the two hemp flours for all of the sensory attributes except bulkiness. In general, the use of Hemp 2, with a similar particle size to semolina, to enrich pasta yielded better results than Hemp 1. The absence of proteins such as glutenins and gliadins, responsible for the formation of gluten, has inevitably influenced the characteristics of the product. CTRL recorded the greatest OQS, mainly in terms of firmness and adhesiveness (Table 9); the enrichment of durum wheat pasta with non-gluten flours may affect the parameters, resulting in an increase in adhesiveness. The odor and taste of pasta fortified at different substitution levels resulted similarly to CTRL pasta, suggesting that the particle size of hemp did not affect these traits. The best results in terms of OQS were obtained using Hemp 2 at 7.5% substitution. 

As regards to cooking quality, the replacement of durum wheat semolina with both types of hemp flours in the pasta statistically influenced (*p* < 0.05) the water absorption, most probably due to the high dietary fiber content and resultant strong water absorption capacity [13,15], while the optimal cooking time compared to the CTRL sample increased, particularly when using the Hemp 1 flours (Table 10).

The amount of solid substance lost in the cooking water (cooking loss) did not result as being influenced by the hemp particle size or the percentage of substitution, meaning that the hemp flours were well retained in the pasta. According to Sicignano et al., [31] the hydration of semolina with a wide range of particle sizes affects dough development and pasta quality; the different particle sizes between the semolina and Hemp 1 flours probably led to an over-hydration of the finer fraction and under-hydration of the coarser fraction, affecting the WA and OCT (Table 10). 

## 4. Discussion

The main aim of this study was to develop pasta fortified with variable percentages of hemp flour with different particle sizes. Our results revealed that the incorporation of hemp flour into the pasta formulas led to significant increases in the TPC and DPPH values, AAs, FA composition, and sensory qualities, in addition to obtain satisfactory properties and good cooking qualities, related to the percentage of hemp substitution used in the production recipe. The fibrous sensation recorded was probably due to the different particle sizes between the semolina and Hemp 1 flour and to the different percentages of water required for the dough development detected by farinograph analysis. Blends of semolina and Hemp 2 required less water (on average 58%) vs Hemp 1 (on average 61%) (data not shown).

The Folin–Ciocalteau method was used for raw material; similar polyphenol content values were obtained for Hemp 1 and Hemp 2 (6.38 and 6.35 mg GAE/g, respectively), while the TPC value was lower for Ciclope flour (2.45 mg GAE/g). TPC values increased with the addition of hemp flour in pasta. The highest increase in TPC was observed in pasta samples containing 10% hemp flour (4.92 ± 0.31 mg/GAE and 4.21 ± 0.35 mg/GAE for Hemp 1 and Hemp 2, respectively) while the lowest was recorded for the CTRL pasta sample (1.11 ± 0.18 mg/GAE).

However, despite the loss in amino acids and phenolics during cooking (about 40%), the enriched pasta still showed good antioxidant activity. The improvements to DPPH values were found to be higher in pasta formulas with the addition of hemp flour than in CTRL samples prepared with 100% Ciclope wheat flour. In fact, the supplementation of 10% hemp flour also enhanced the antioxidant activity (3.86 ± 0.07 mmol TE/100 g and 3.14 ± 0.06 mmol TE/100 g for Hemp 1 and Hemp 2 pasta samples, respectively) compared to the CTRL samples (1.14 ± 0.05 mmol TE/100 g).

This study also focused on the AAs composition of fortified pasta. The contents of some amino acids considered essential in the human diet can be low in wheat proteins, especially lysine and threonine. The preparation of a functional pasta enriched with variable percentages of hemp flour could affect the content of these two amino acids.

From observed data, the lysine content was found to be 0.16, 0.85–0.36 g/100 g in the CTRL, Hemp 1_10, and Hemp 2_10 flours respectively, while threonine was found to be 0.38, 0.72–1.17 g/100 g in the CTRL, Hemp 1_10, and Hemp 2_10 flours respectively). The total essential amino acids in the fortified pasta samples were 4.30 and 4.62 g/100 g (in Hemp 1_10 and in Hemp 2_10, respectively), while in the durum wheat pasta they were 2.05 g/100 g.

An increase in mono- and polyunsaturated fatty acids was also observed in fortified pasta. The total ω 3 contents varied between 15.9 and 15.3% in Hemp 2 pasta samples and between 15.9–14.7% in Hemp 1 pasta samples. Linoleic was the fatty acid present in the largest amount in Hemp 2 pasta samples (47.94%). The control sample had a lower amount of total ω 3 (4.02%).

Phenols represent the most relevant compounds found in hemp, including some phenylamides, phenolic acids, lignanamides, and flavonoids, such as flavonols, flavones, and flavanols. Current literature suggests that the long-term consumption of diets rich in phenolic compounds protects against certain cancers, cardiovascular diseases, type 2 diabetes, osteoporosis, lung damage, and neurodegenerative diseases [32,33].

An untargeted metabolomics-based approach was used to comprehensively screen and profile phenols in different hemp flours and pasta samples through UHPLC-ESI/QTOF-MS analysis. A total of 12 phenols were identified (Table 8) by comparison with the retention time, MS spectra, and accurate mass measurement obtained from the literature data [17,18,19,20,21] and by phenolic reference standards (Appendix A).

As is known, phenolic compounds are contained in plant materials in the free (PCs) but also in the insoluble bound form (BPs); these latter are covalently linked to the structural components of the cell wall. Particularly interesting are the implications of BPs in foods, in terms of bioaccessibility, transformation during digestion, and modulation of the gut microbiota [34]. For this reason, the study of the bound polyphenolic fraction (BPs) in pasta samples was of interest. 

In raw samples, the contents of free phenolic compounds (PCs) were rather low in Ciclope flour; hydroxybenzoic and protocatechuic acids represented the most abundant phenolic compounds quantified in hemp flour samples. The results of the phenolic component after alkaline treatment of the matrices showed a higher amount of these in flour. Regarding fortified pastas, the phenol content is good, especially in the bound form. As could easily be predicted, the content of these increases with the percentage of replacement. Cannabisin B was only found in Hemp 1 as a bound phenol and at lower levels with respect to cannaflavin C. Cannaflavin A, cannaflavin B, and cannabisins A, B, and C are non-psychoactive molecules exclusively present in hemp plants that suppress PGE2 production in synovial membrane cells, exhibiting anti-inflammatory power 30-times stronger than acetylsalicylic acid [35]. Among hemp’s exclusive lignanamides, these exhibit remarkable beneficial effects on human health [36]. 

Caffeoyltyramine and its phenolic amides, including cis-N-caffeoyltyramine and trans-N-caffeoyltyramin, are known to have anti-fungal, antioxidant, anti-inflammatory, and antihyperlipidemic activities [28]. As observed from Table 8, the phenols cannaflavin C, *p*-hydroxybenzoic acid, protocatechuic acid, hydroxycinnamic acid, and caffeic acid were predominantly found in Hemp 1 pasta in bound form.

In the pasta samples obtained by replacing the semolina with Hemp 2 flour, *p*-hydroxycinnamic acid, caffeic acid, *p*-hydroxybenzoic acid, protocatechuic acid, and trans-N-caffeoyltyramine always resulted to a greater extent in the bound form. As expected, the amount of free and bound phenols is strongly related to the fortification rate.

The incorporation of hemp flours to produce pasta has been studied by other research teams, but the contribution of mineral content was not investigated. Our results represent the first finding of the addition of hemp flours producing fortified pasta rich in minerals. However, in this study the presence of phytates in hemp flours was not detected, and even if the pasta samples could be an excellent source of mineral elements, their nutritional quality could be reduced.

Each increase in the addition of hemp flour resulted in an increase in pasta quality (water absorption, due to gluten dilution) and in the satisfactory organoleptic properties (until 7.5%); the fibrous sensation recorded at higher percentages of substitution was probably due to the different particle sizes between semolina and Hemp 1 flour and to the different percentages of water required for the dough development detected by farinograph analysis. Blends of semolina and Hemp 2 required less water (on average, doses of 58%) compared to Hemp 1 (on average, doses of 61%). Additionally, the contribution of hemp enrichment improved the micro-elemental composition of pasta (iron, potassium, magnesium, and phosphorus) compared to the control sample. The number of health claims relating to mineral elements which could be used for hemp flours could be high, but further studies are needed on the bioaccessibility and bioavailability in order to clarify the role of hemp flour as a dietary source of mineral elements due to the presence of phytates in the raw material.

## 5. Conclusions

The present study concluded that the incorporation of hemp flours with different particle sizes represents the best compromise between pasta properties and nutrient content; the addition of hemp flour to durum semolina cv Ciclope flour effectively increased the anti-radical potential. The best results in terms of overall quality score (OQS) were obtained using flour with a minor particle size (Hemp 2) for the preparation of the pasta. The maximum substitution level, which showed the best performance during the production process, was 7.5% for both types of hemp flour used. However, some differences can be highlighted: specifically, the pasta obtained by the replacement of 7.5% of the Ciclope semolina with Hemp 2 flour showed a better profile in terms of mineral salts and amino acids and a greater quantity of polyunsaturated fatty acids, while the pasta obtained from the same percentage of substitution but with Hemp 1 flour showed a better phenolic profile and TCP and better anti-radical activity. Hemp supplementation could be a potential option for the production of high-quality, nutritionally rich, low-cost pasta with good organoleptic properties. 

In summary, the enrichment of wheat pasta with hemp flours is a very interesting future trend that enables more attractive pasta products to be obtained for consumers in terms of increased nutritional and pro-health value. On the basis of these results, hemp-enriched pasta may have great potential in the industry for the development of functional products. Further studies on the bioavailability of nutrients, the glycemic index, and the effects on the intestinal microbiota of Hemp 1- and Hemp 2-fortified pasta will be conducted in the near future.

## Figures and Tables

**Figure 1 foods-12-00774-f001:**
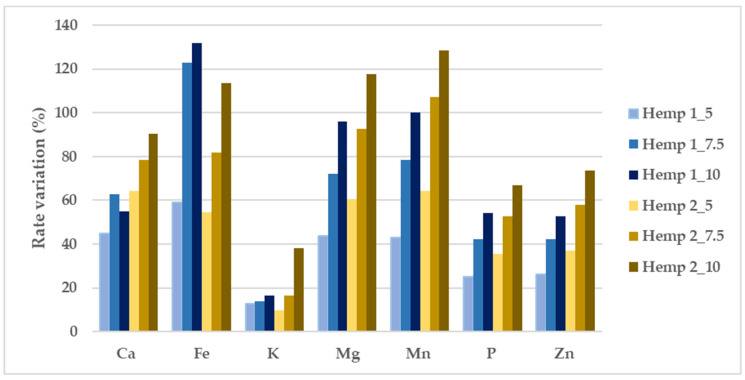
Percentage variation (%) of the hemp-enriched pasta samples vs the CTRL (durum wheat semolina cv. Ciclope).

**Table 1 foods-12-00774-t001:** Ciclope and fortified pasta samples.

Material			Fortified Hemp Pasta		
		0	5%	7.5%	10%
Ciclope semolina	(Ciclope)	CTRL			
Hemp 1 flour	(Hemp 1)		Hemp 1_5	Hemp 1_7.5	Hemp 1_10
Hemp 2 flours	(Hemp 2)		Hemp 2_5	Hemp 2_7.5	Hemp 2_10

Hemp 1 is hemp flour sieved at 0.530 mm; Hemp 2 is flour sieved at 0.236 mm. The fortified pastas take into account the different percentages of substitution of semolina with the two different hemp flours.

**Table 2 foods-12-00774-t002:** Operating conditions and acquisition parameters for ICP-MS.

Parameter	Setting
RF power (W)	500–1700
Reflected power	<10
Plasma gas flow (L min^−1^)	15
Nebulizer gas flow (L min^−1^)	1.00
Auxiliary gas flow (L min^−1^)	0.80
He mode	collision cell mode
He gas flow (mL min^−1^)	5.00
Octopole bias (CCT bias) (V)	−21
Quadrupole bias (pole bias) (V)	−18

**Table 3 foods-12-00774-t003:** Operating conditions and acquisition parameters for ICP-OES.

Parameter	Setting
Nebulizer	Glass concentric
Nebulizer Gas Flow (L min^−1^)	0.40
Spray chamber	Glass Cyclonic
Purge Gas Flow	Normal
Auxiliary gas flow (L min^−1^)	0.50
Coolant gas flow (L min^−1^)	12
RF Power (W)	1150
Pump Speed (rpm)	50

**Table 4 foods-12-00774-t004:** Chemical characterization of raw materials.

	Ciclope	Hemp 1	Hemp 2
TPC mgGAE/g	2.45 ± 0.001	6.38 ± 0.002	6.35 ± 0.001
DPPH_TEAC_ mmol TE/100 g	1.35 ± 0.0355	3.94 ± 0.0178	3.75 ± 0.0179
% Scavenging	29.7	53.2	51.5
Fatty acid (Relative percentage %)			
Palmitic acid	17.6 ± 0.30	9.04 ± 0.17	9.10 ± 0.057
Stearic acid	1.54 ± 0.16	3.42 ± 0.025	3.32 ± 0.060
Oleic acid	17.3 ± 0.51	14.6 ± 0.062	14.5 ± 0.081
Linoleic acid (ω-6)	59.9 ± 0.89	53.7 ± 0.35	53.7 ± 0.17
γ linolenic acid (ω-6)	-	2.44 ± 0.075	2.47 ± 0.089
α linolenic acid (ω-3)	3.62 ± 0.53	15.7 ± 0.21	15.5 ± 0.050
∑ ω-6	59.9 ± 0.89	56.1 ± 0.042	56.1 ± 0.259
∑ ω-3	3.62 ±0.53	15.7 ± 0.21	15.5 ± 0.050
Amino acids (g/100 g)			
Arginine	0.34 ± 0.016	0.52 ± 0.020	0.58 ± 0.018
Serine	1.59 ± 0.035	2.00 ± 0.041	1.89 ± 0.038
Glutamine	2.90 ± 0.031	4.96 ± 0.039 b	5.38 ± 0.041 a
Tyrosine	3.37 ± 0.016	3.43 ± 0.014	3.19 ± 0.012
Alanine	0.85 ± 0.009	1.37 ± 0.011	1.84 ± 0.012
Histidine	0.53 ± 0.002	0.92 ± 0.002	0.94 ± 0.003
Proline	1.23 ± 0.025	3.22 ± 0.051	2.99 ± 0.057
Threonine	1.80 ± 0.015	1.97 ± 0.019 b	2.51 ± 0.021 a
Leucine	1.78 ± 0.035	2.75 ± 0.036	2.97 ± 0.035
Methionine	0.14 ± 0.006	0.61 ± 0.011	0.83 ± 0.016
Valine	0.58 ± 0.005	1.17 ± 0.012 b	2.00 ± 0.045 a
Phenylalanine	0.25 ± 0.015	0.40 ± 0.012	0.45 ± 0.013
Isoleucine	3.52 ± 0.055	6.94 ± 0.057	6.21 ± 0.053
Lysine	0.16 ± 0.005	2.07 ± 0.036 b	2.58 ± 0.012 a
∑ AA	19.0 ± 2.39	32.3 ± 3.61 b	34.4 ± 3.76 a
∑ essential AA	8.23 ± 1.03	15.9 b ± 1.78	17.6 ± 1.81 a

Data are expressed as the means ± SD of triplicate experiments. Hemp 1 and Hemp 2 were diluted 10-fold for anti-scavenger determination. Different letters indicated differences at *p <* 0.05 between Hemp 1 and Hemp 2 samples. Values without letters are not significantly different.

**Table 5 foods-12-00774-t005:** Phenolic compounds (µg 100 g^−1^): Free and bound quantification (PCs and BPs) of raw materials.

	Ciclope		Hemp 1		Hemp 2	
	PCs	BPs	PCs	BPs	PCs	BPs
*p*-Hydroxybenzoic acid	n.d.	n.d.	110.2 ± 1.09	866.0 ± 2.21	73.00 ± 1.39	542.3 ± 2.03
Protocatechuic acid	n.d.	189.5 ± 1.53	45.10 ± 1.48	1210 ± 3.60	79.10 ± 1.87	1361 ± 2.43
Hydroxycinnamic acid	n.d.	329.6 ± 1.33	n.d.	1687 ± 2.02	82.00 ± 1.74	1589 ± 2.69
Vanillic acid	n.d..	75.20 ± 1.37	n.d.	.	n.d.	710.7 ± 1.65
Caffeic acid	n.d.	n.d.	n.d.	849.0 ± 1.23	n.d.	1185 ± 1.97
Ferulic acid	n.d.	n.d.	n.d.	616.2 ± 1.40	30.10 ± 1.11	243.6 ± 1.21
Sinapic acid	n.d.	n.d.	n.d.	550.4 ± 1.23	n.d.	476.5 ± 1.01
Catechin	n.d.	n.d.	n.d.	407.3 ± 0.98	n.d.	n.d.
N-trans-Caffeoyltyramine	n.d.	n.d.	n.d.	n.d.	n.d.	1817 ± 1.42
Chlorogenic acid	n.d.	n.d.	n.d.	353.7 ± 0.89	n.d.	n.d.
Cannabisin B	n.d.	n.d.	n.d.	538.3 ± 0.85	n.d.	n.d.
Cannaflavin C	n.d.	n.d.	1384 ± 1.77	3367 ± 1.15	1139 ± 0.98	2207 ± 1.45

Data are expressed as means ± SD of triplicate experiments. n.d. not detected.

**Table 6 foods-12-00774-t006:** Chemical characterization of cooked pasta.

	CTRL	Hemp 1_5	Hemp 1_7.5	Hemp 1_10	Hemp 2_5	Hemp 2_7.5	Hemp 2_10	LSD (*p* < 0.05)
TPC mgGAE/g	1.11 ± 0.18	2.50 ± 0.37	4.25 ± 0.18	4.92 ± 0.31	1.95 ± 0.33	2.76 ± 0.40	4.21 ± 0.35	0.31
DPPH_TEAC_ mmol TE/100 g	1.14 ± 0.05	2.30 ± 0.04	3.08 ± 0.08	3.86 ± 0.07	2.08 ± 0.08	2.65 ± 0.09	3.14 ± 0.06	0.08
% Scavenging	27.8	38.3	45.4	52.5	36.4	41.6	45.9	0.08
Fatty acid (Relative percentages%)								
Palmitic acid	15.92 ± 1.09	14.72 ± 0.4	15.11 ± 1.04	15.56 ± 0.3	15.33 ± 0.48	16.40 ± 0.8	16.64 ± 0.89	1.44
Stearic acid	2.72 ± 0.81	1.54 ± 0.26	2.73 ± 0.31	3.13 ± 0.8	3.29 ± 0.34	3.66 ± 0.66	3.88 ± 0.89	0.92
Oleic acid	12.96 ± 0.53	12.30 ± 1.6	13.80 ± 1.4	15.68 ± 0.9	13.42 ± 0.23	15.29 ± 0.46	17.81 ± 0.7	0.97
Linoleic acid	46.40 ± 0.62	41.66 ± 0.87	42.16 ± 0.8	43.05 ± 1.6	46.36 ± 1.23	47.26 ± 0.16	47.94 ± 0.61	0.91
γ linolenic acid	0.86 ± 0.075	3.37 ± 0.30	4.33 ± 0.8	5.26 ± 0.11	4.84 ± 0.04	4.87 ± 0.19	5.71 ± 0.40	1.27
α linolenic acid	4.02 ± 0.63	14.70 ± 0.4	15.51 ± 1.04	15.89 ± 0.3	15.38 ± 0.48	15.61 ± 0.8	15.89 ± 0.86	0.68
∑ ω 6	47.3	44.9	46.5	48.3	51.2	52.1	53.6	
∑ ω 3	4.02	14.7	15.5	15.9	15.3	15.6	15.9	
Amino acids (g/100 g)								
Arginine	0.14 ± 0.015	0.17 ± 0.020	0.19 ± 0.006	0.22 ± 0.012	0.19 ± 0.012	0.23 ± 0.020	0.26 ± 0.015	0.03
Serine	0.44 ± 0.015	0.42 ± 0.010	0.52 ± 0.021	0.68 ± 0.012	0.57 ± 0.015	0.67 ± 0.026	0.86 ± 0.015	0.03
Glutamine	0.74 ± 0.020	1.13 ± 0.015	1.19 ± 0.012	1.33 ± 0.020	1.12 ± 0.015	1.24 ± 0.015	1.33 ± 0.021	0.03
Tyrosine	0.44 ± 0.025	0.81 ± 0.015	1.17 ± 0.025	1.53 ± 0.020	0.95 ± 0.021	1.21 ± 0.015	1.49 ± 0.015	0.04
Alanine	0.22 ± 0.020	0.29 ± 0.015	0.32 ± 0.010	0.35 ± 0.015	0.39 ± 0.010	0.43 ± 0.017	0.48 ± 0.015	0.02
Histidine	0.1 ± 0.015	0.20 ± 0.015	0.36 ± 0.015	0.45 ± 0.015	0.26 ± 0.015	0.33 ± 0.020	0.53 ± 0.015	0.03
Proline	0.68 ± 0.010	1.27 ± 0.015	1.43 ± 0.021	1.79 ± 0.021	1.41 ± 0.020	1.65 ± 0.012	1.80 ± 0.021	0.03
Threonine	0.38 ± 0.021	0.49 ± 0.010	0.59 ± 0.015	0.72 ± 0.020	0.90 ± 0.017	1.03 ± 0.021	1.17 ± 0.015	0.03
Leucine	0.16 ± 0.015	0.40 ± 0.015	0.45 ± 0.006	0.57 ± 0.020	0.50 ± 0.021	0.62 ± 0.025	0.69 ± 0.015	0.03
Methionine	0.08 ± 0.001	0.11 ± 0.010	0.15 ± 0.020	0.19 ± 0.010	0.13 ± 0.012	0.16 ± 0.015	0.18 ± 0.015	0.02
Valine	0.09 ± 0.002	0.12 ± 0.010	0.23 ± 0.020	0.30 ± 0.025	0.29 ± 0.010	0.33 ± 0.015	0.37 ± 0.015	0.02
Phenylalanine	0.18 ± 0.012	0.15 ± 0.008	0.20 ± 0.015	0.26 ± 0.012	0.25 ± 0.020	0.29 ± 0.010	0.33 ± 0.015	0.02
Isoleucine	1.00 ± 0.021	1.08 ± 0.020	1.21 ± 0.015	1.41 ± 0.025	1.24 ± 0.015	1.36 ± 0.020	1.52 ± 0.015	0.03
Lysine	0.16 ± 0.020	0.03 ± 0.001	0.52 ± 0.010	0.85 ± 0.020	0.08 ± 0.003	0.22 ± 0.020	0.36 ± 0.021	0.02
∑ AA	4.81 ± 0.070	6.67 ± 0.052	8.53 ± 0.110	10.65 ± 0.045	8.28 ± 0.084	9.77 ± 0.085	11.37 ± 0.035	0.12
∑ essential AA	2.05 ± 0.049	2.38 ± 0.027	3.35 ± 0.055	4.30 ± 0.050	3.39 ± 0.048	4.01 ± 0.081	4.62 ± 0.036	0.09

Data are expressed as means ± SD of triplicate experiments. Means were separated by LSD test at *p* < 0.05.

**Table 7 foods-12-00774-t007:** Phenolic compounds (µg 100 g^−1^): free and bound quantification in cooked pasta (PCs and BPs).

	Hemp 1_5		Hemp 2_5		Hemp 1_7.5		Hemp 2_7.5		Hemp 1_10		Hemp 2_10	
	PCs	BPs	PCs	BPs	PCs	BPs	PCs	BPs	PCs	BPs	PCs	BPs
*p*-Hydroxybenzoic acid	n.d.	n.d.	n.d.	164.0 ± 1.02	n.d.	n.d.	n.d.	342.5 ± 1.43	n.d.	496.0 ± 2.01	n.d.	444.2 ± 2.09
Protocatechuic acid	n.d.	320.1 ± 0.89	n.d.	n.d.	n.d.	453.5 ± 1.43	n.d.	n.d.	n.d.	761.0 ± 1.87	n.d.	446.2 ± 1.25
Hydroxycinnamic acid	n.d.	410.3 ± 1.24	n.d.	219.9	n.d.	764.2 ± 1.29	n.d.	802.9 ± 1.98	n.d.	1033 ± 1.78	n.d.	571.3 ± 1.56
Vanillic acid	n.d.	110.0 ± 1.47	n.d.	n.d.	n.d.	335.1 ± 1.06	n.d.	n.d.	n.d.	385.4 ± 1.52	n.d.	n.d.
Caffeic acid	n.d.	n.d.	n.d.	107.9 ± 0.84	n.d.	203.4 ± 1.16	n.d.	367.6 ± 1.26	n.d.	519.3 ± 1.06	n.d.	846.5 ± 1.97
Ferulic acid	n.d.	n.d.	n.d.	n.d.	n.d.	289.4 ± 1.44	n.d.	n.d.	n.d.	372.3 ± 099	n.d.	n.d.
Sinapic acid	n.d.	n.d.	n.d.	67.9 ± 0.89	n.d.	216.4 ± 1.79	n.d.	295.6 ± 1.09	n.d.	410.7 ± 1.32	n.d.	313.4 ± 1.19
Catechin	n.d.	n.d.	n.d.	n.d.	n.d.	n.d.	n.d.	n.d.	n.d.	n.d.	n.d.	n.d.
N-trans-Caffeoyltyramine	n.d.	n.d.	n.d.	n.d.	n.d.	n.d.	n.d.	n.d.	n.d.	n.d.	n.d.	307.3 ± 1.96
Chlorogenic acid	n.d.	n.d.	n.d.	n.d.	n.d.	n.d.	n.d.	n.d.	n.d.	n.d.	n.d.	n.d.
Cannabisin B	n.d.	n.d.	n.d.	n.d.	n.d.	n.d.	n.d.	n.d.	n.d.	n.d.	n.d.	n.d.
Cannaflavin C	132.9 ± 1.45	103.1 ± 1.23	115.9 ± 1.03	110.1 ± 1.78	279.6 ± 1.02	587.4 ± 1.24	176.5 ± 1.80	101.0 ± 1.09	382.0 ± 1.07	855.2 ± 1.69	227.8 ± 1.70	274.8 ± 1.53

Data are expressed as the means ± SD of triplicate experiments. n.d., not detected.

**Table 8 foods-12-00774-t008:** Mineral content in cooked, fortified pasta.

mg g^−1^	CTRL	Hemp 1_5	Hemp 1_7.5	Hemp 1_10	Hemp 2_5	Hemp 2_7.5	Hemp 2_10
Al	0.005 ± 0.000	0.034 ± 0.005	0.017 ± 0.000	0.036 ± 0.014	0.012 ± 0.002	0.025 ± 0.004	0.013 ± 0.005
B	<0.002	0.002 ± 0.000	0.002 ± 0.000	0.002 ± 0.000	0.002 ± 0.000	0.002 ± 0.000	0.003 ± 0.000
Ca	0.401 ± 0.0024	0.581 ± 0.021	0.653 ± 0.075	0.621 ± 0.014	0.659 ± 0.067	0.716 ± 0.068	0.764 ± 0.040
Fe	0.022 ± 0.000	0.035 ± 0.001	0.049 ± 0.004	0.051 ± 0.002	0.034 ± 0.001	0.040 ± 0.001	0.047 ± 0.004
K	1.735 ± 0.005	1.953 ± 0.005	1.976 ± 0.024	2.020 ± 0.025	1.907 ± 0.005	2.019 ± 0.005	2.398 ± 0.014
Mg	0.608 ± 0.074	0.873 ± 0.111	1.047 ± 0.130	1.191 ± 0.156	0.976 ± 0.130	1.172 ± 0.156	1.323 ± 0.153
Mn	0.014 ± 0.000	0.020 ± 0.000	0.025 ± 0.001	0.028 ± 0.001	0.023 ± 0.001	0.029 ± 0.000	0.032 ± 0.001
Na	0.092 ± 0.012	0.101 ± 0.015	0.095 ± 0.016	0.101 ± 0.014	0.070 ± 0.011	0.097 ± 0.015	0.087 ± 0.013
P	2.000 ± 0.023	2.502 ± 0.013	2.845 ± 0.000	3.086 ± 0.040	2.707 ± 0.024	3.054 ± 0.063	3.337 ± 0.036
Zn ug g^−1^	0.019 ± 0.001	0.024 ± 0.001	0.027 ± 0.000	0.029 ± 0.001	0.026 ± 0.001	0.030 ± 0.001	0.033 ± 0.001
As	0.006 ± 0.000	0.007 ± 0.000	0.007 ± 0.000	0.008 ± 0.000	0.004 ± 0.000	0.006 ± 0.000	0.006 ± 0.000
Ba	0.777 ± 0.003	1.000 ± 0.003	1.079 ± 0.004	1.276 ± 0.003	0.910 ± 0.003	1.058 ± 0.004	1.009 ± 0.003
Be	<0.01	<0.01	<0.01	<0.01	<0.01	<0.01	<0.01
Cd	0.024 ± 0.000	0.022 ± 0.000	0.022 ± 0.000	0.024 ± 0.000	0.024 ± 0.000	0.025 ± 0.000	0.028 ± 0.000
Co	0.090 ± 0.000	0.094 ± 0.000	0.095 ± 0.000	0.103 ± 0.000	0.099 ± 0.000	0.105 ± 0.000	0.114 ± 0.000
Cr	0.036 ± 0.000	0.039 ± 0.000	0.059 ± 0.000	0.074 ± 0.000	0.035 ± 0.000	0.065 ± 0.000	0.060 ± 0.000
Cu	6.905 ± 0.024	7.903 ± 0.028	8.266 ± 0.029	9.669 ± 0.019	8.320 ± 0.025	9.063 ± 0.032	10.217 ± 0.025
Hg	0.005 ± 0.000	0.004 ± 0.000	0.007 ± 0.000	0.004 ± 0.000	0.004 ± 0.000	0.005 ± 0.000	0.003 ± 0.000
Li	0.054 ± 0.00	0.050 ± 0.00	0.054 ± 0.00	0.085 ± 0.00	0.042 ± 0.00	0.046 ± 0.00	0.062 ± 0.00
Mo	0.490 ± 0.002	0.572 ± 0.002	0.578 ± 0.002	0.626 ± 0.001	0.622 ± 0.002	0.683 ± 0.002	0.760 ± 0.002
Ni	0.279 ± 0.001	0.414 ± 0.001	0.448 ± 0.002	0.566 ± 0.001	0.482 ± 0.001	0.608 ± 0.002	0.763 ± 0.002
Pb	0.552 ± 0.002	0.533 ± 0.002	0.550 ± 0.002	0.532 ± 0.001	0.519 ± 0.002	0.526 ± 0.002	0.575 ± 0.001
Sb	0.004 ± 0.000	0.011 ± 0.000	0.006 ± 0.000	0.003 ± 0.000	0.004 ± 0.000	0.003 ± 0.000	0.006 ± 0.000
Se	0.034 ± 0.000	0.044 ± 0.000	0.046 ± 0.000	0.058 ± 0.000	0.038 ± 0.000	0.039 ± 0.000	0.048 ± 0.000
Sn	0.003 ± 0.000	0.006 ± 0.000	0.014 ± 0.000	0.007 ± 0.000	0.018 ± 0.000	0.017 ± 0.000	0.026 ± 0.000
Sr	4.095 ± 0.014	5.832 ± 0.020	7.334 ± 0.026	8.525 ± 0.017	6.709 ± 0.020	8.808 ± 0.031	8.600 ± 0.021
V	0.016 ± 0.000	0.058 ± 0.000	0.064 ± 0.000	0.086 ± 0.000	0.021 ± 0.000	0.046 ± 0.000	0.045 ± 0.000

Data are expressed as the means ± SD of triplicate experiments.

**Table 9 foods-12-00774-t009:** Sensory characteristics of cooked pasta.

Sample	Bulkiness ^I^	Firmness ^I^	Adhesiveness ^I^	Fibrous ^I^	Color ^II^	Odor ^III^	Taste ^III^	OQS ^III^
CTRL	3.7 ± 0.012	5.7 ± 0.018	4.3 ± 0.014	5.0 ± 0.016	5.0 ± 0.016	4.7 ± 0.015	6.3 ± 0.021	6.0 ± 0.020
Hemp 1_5	3.0 ± 0.010	4.7 ± 0.015	5.0 ± 0.016	4.7 ± 0.015	4.7 ± 0.015	4.7 ± 0.015	5.3 ± 0.017	5.3 ± 0.017
Hemp 1_7.5	3.7 ± 0.012	4.0 ± 0.016	4.7 ± 0.015	5.7 ± 0.018	5.0 ± 0.016	5.7 ± 0.018	5.7 ± 0.018	5.3 ± 0.017
Hemp 1_10	4.5 ± 0.015	5.0 ± 0.013	3.5 ± 0.011	4.5 ± 0.015	5.0 ± 0.016	4.5 ± 0.015	6.0 ± 0.020	4.0 ± 0.013
Hemp 2_5	3.7 ± 0.012	4.0 ± 0.013	4.3 ± 0.014	5.0 ± 0.016	5.3 ± 0.017	5.3 ± 0.017	5.3 ± 0.017	5.3 ± 0.017
Hemp 2_7.5	5.3 ± 0.017	5.7 ± 0.018	6.0 ± 0.020	5.7 ± 0.018	6.3 ± 0.021	5.7 ± 0.018	6.7 ± 0.022	6.3 ± 0.021
Hemp 2_10	4.3 ± 0.014	5.0 ± 0.016	4.7 ± 0.015	5.7 ± 0.018	5.0 ± 0.016	4.7 ± 0.015	5.3 ± 0.017	5.0 ± 0.016
LSD _(*p* < 0.05)_	0.034	0.039	0.036	0.040	0.041	0.039	0.045	0.042
Percentage of substitution (S)								
CTRL	3.7 ± 0.012 b	5.7 ± 0.018 a	4.3 ± 0.014 c	5.0 ± 0.016 b	5.0 ± 0.016 b	4.7 ± 0.015 c	6.3 ± 0.021 b	6.0 ± 0.020 a
5%	3.3 ± 0.013 b	4.3 ± 0.014 b	4.6 ± 0.015 b	4.8 ± 0.015 c	5.0 ± 0.016 b	5.0 ± 0.016 b	5.3 ± 0.017 d	5.3 ± 0.017 b
7.5%	4.5 ± 0.015 a	4.8 ± 0.017 a	5.3 ± 0.018 a	5.6 ± 0.018 a	5.6 ± 0.019 a	5.7 ± 0.018 a	6.7 ± 0.020 a	5.8 ± 0.024 a
10%	4.4 ± 0.015 a	5.0 ± 0.015 a	4.0 ± 0.013 c	5.1 ± 0.017 b	5.0 ± 0.016 b	4.6 ± 0.015 c	5.7 ± 0.019 c	4.5 ± 0.015 c
Particle size (PS)								
Hemp 1	3.7 ± 0.012 a	4.5 ± 0.015 a	4.4 ± 0.014 b	4.9 ± 0.016 b	4.9 ± 0.016 b	4.9 ± 0.016 a	5.7 ± 0.018 a	4.9 ± 0.016 b
Hemp 2	4.4 ± 0.014 a	4.9 ± 0.016 b	4.8 ± 0.016 a	5.4 ± 0.018 a	5.5 ± 0.018 a	5.2 ± 0.017 a	5.8 ± 0.019 a	5.5 ± 0.018 a
ANOVA								
Main effects								
S	***	***	***	***	***	**	***	***
PS	***	***	***	ns	***	ns	ns	***
Interaction								
S × PS	***	***	***	***	***	*	***	***

Data are expressed as the means ± SD of triplicate experiments. ***; **, and * indicate significance at *p* < 0.001, *p* < 0.01, and *p* < 0.05, respectively, while “ns” is not significant. The differences among all samples were detected by LSD test at *p* < 0.05. Different small letters in a column for “PS” and “S” factors indicate statistical differences among samples (*p <* 0.05) (Student–Newman–Keuls test). OQS means “over quality score”. I: 1—low sensation, -9—high sensation; II: 1—very clear, -9—very dark; III: 1—extremely unpleasant, -9—extremely pleasant; n = 24.

**Table 10 foods-12-00774-t010:** The cooking quality, optimum cooking time (OCT), water absorption capacity (WA), and cooking loss of fortified pasta samples.

	OCT (min)	WA	Cooking Loss
CTRL	12.0	75 ± 1.3	0.99 ± 0.001
Hemp 1_5	13.0 a	100 ± 2.8 a	0.99 ± 0.001
Hemp 2_5	11.5 b	75 ± 1.6 b	1.00 ± 0.001
Hemp 1_7.5	12.5 a	125 ± 3.2 a	0.99 ± 0.001
Hemp 2_7.5	12.0 a	125 ± 3.0 a	0.99 ± 0.001
Hemp 1_10	12.5 a	150 ± 3.9 a	0.99 ± 0.001
Hemp 2_10	12.5 a	100 ± 2.4 b	0.99± 0.001

Different small letters in a column. indicate statistical differences between the hemp flours used for the same concentration at *p <* 0.05; n = 3. Data are expressed as the means ± SD of triplicate experiments.

## Data Availability

The data presented in this study are available on request from the corresponding author.

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
