# Peer review of "Hemp Flour Particle Size Affects the Quality and Nutritional Profile of the Enriched Functional Pasta"

_foods, 2023, doi:10.3390/foods12040774_

Round 1

Author Response

The revised version of the manuscript is more comprehensive and certainly meets journal style requirements, according to manuscript format and references. Authors have accepted many of the reviewers; recommendations and addressed to most of their suggestions successfully.

Thanks to Reviewer #1 for his/her positive comments on the quality improvement of the manuscript.

REV#1:

Dear Authors, the article is interesting and well written, being an interest topic for the actual context of food security.

This research studied the rheological and chemical qualities of fortified pasta obtained by using the durum wheat cultivar "Ciclope", fortified with different percentages of hemp flour (cv. Futura 75) to develop a fortified pasta and evaluated the effect of particle size of hemp flours on sensory properties, cooking quality and nutritional characteristics of functional pasta.

 Anyway, considering also the field of research of this journal the work is very valuable and I would like to congratulate the authors for their work.

The introduction section is very well documented and written.

Anyway, at Line 96 the aim of this study does not appear well defined.

The methods are well described and can be easly reproduced. The authors strictly followed all analysis methods.

The experimental part is accurately done and very complex, and the results are well-presented bringing solid discussions.

Also, the Conclusion Section is very well summarised and highlights the results of the study.

A: The authors improved the description of the study objectives by changing the sentence.

Thanks to Reviewer #1 for his/her positive opinion on the topic of the paper and his/her precious comments as well. The authors agree with the proposed suggestion. Thank you very much for supporting the publication of the manuscript on Foods.

Reviewer 2 Report

1.      Abstract has to revise, all the methods can be in one place, and then resent results and conclusions

2.      Try to give a few lines about the rheological properties of pasta, and how can effect by composite flour usage.

3.      The second paragraph on page 2 can be split into different paragraphs

4.      Section 2.1 can be removed

5.      In section 2.2 heap flour and hemp seed brought separately? Check once

6.      Table 1, give the footnote properly by mentioning what is Hemp1,2 etc..   

7.      Line 14, what is OCT?

8.      Why can separate, sections into pasta making, cooking quality, and sensory evaluation in different sections?

9.      In section 2.4, what is pasta? Is it raw or boiled?

10.  Give the footnote of Table 4, what is Hemp 1_5, Hemp 1_7.5

11.  Table 5, what is OCT? and WA has to be given in the footnote?

12.  Section3.1 Must discuss well. Need to give clarity on how the sensory quality is affected and why?

13.  Must explain clearly OCT, WA, Cooking loss

14.  In section 3.2 give the subsections of 3.2.1 Total phenolic content, 2.3.2. fatty acids composition etc..

15.  The discussion is a must for section 3.2. The total fatty acids and their health issues should be discussed and mentioned why there is variation among them.

16.  The Amino acid composition is very important, and must be present as the adding the hemp what is the amino acid increased and decreased? What is the nutritional and health influence?

17.  Need to explain the phenolic acids and their health benefits as the incorporation of Hemp. Hence rewrite

18.  The discussion on minerals is very poor.

19.  The section 4, the discussion should elaborate much

20.  Fine-tune the conclusions

21.  Check for the English quality

22.  Check for the typographical and editorial issues 

Author Response

Thanks to Reviewer #2 for their positive comments on the quality improvement of the manuscript.

REV #2

Comments and Suggestions for Authors

  1. Abstract has to revise, all the methods can be in one place, and then resent results and conclusions

A: The abstract has been revised; given the number of analyzes performed, the methods have been listed using the subparagraphs (in paragraph 2.4 we described all the analytical methods).

Results and discussions have also been revised and improved.

  1. Try to give a few lines about the rheological properties of pasta, and how can effect by composite flour usage.

A: Results: in paragraph 3.1 the rheological properties of the pasta have been better described

  1. The second paragraph on page 2 can be split into different paragraphs

A: done

  1. Section 2.1 can be removed

A: Although the authors consider “chemicals” paragraph extremely useful for the reproduction of the experiments, the paragraph 2.1 has been removed.

  1. In section 2.2 heap flour and hemp seed brought separately? Check once

A: in “Raw material “ section we have mentioned and described our starting materials which are Ciclope durum wheat flour and flours obtained from defatted hemp seeds.

  1. Table 1, give the footnote properly by mentioning what is Hemp1,2 etc..

A: authors added as foot note:

Hemp 1 is hemp flour sieved at 0.530 mm; Hemp 2 is flour sieved to 0.236 mm. The fortified pastas take into account the different percentages of substitution of semolina with the two different hemp flours.

  1. Line 14, what is OCT?

A: in lines 134-136 describe the meaning of OTC.

  1. Why can separate, sections into pasta making, cooking quality, and sensory evaluation in different sections?

A: We prefer to leave one section because a lot of descriptions referred to our previous work, where the sections were deeply described and separated.

  1. In section 2.4, what is pasta? Is it raw or boiled?

A: In lines 160 -162 the use of cooked and ground pasta is clearly described.

…..Three replicates (8 g) of each flour sample (Ciclope, Hemp 1 and Hemp 2 flours and ground cooked pasta samples) were homogenized for 45 min in 40 mL 80% methanol  solution using an ultrasonic bath. …..

  1. Give the footnote of Table 4, what is Hemp 1_5, Hemp 1_7.5

A: we wrote it in table 1, I think it is superfluous to write it in all tables.

  1. Table 5, what is OCT? and WA has to be given in the footnote                  A: we changed foot note

Table 5. Cooking quality, optimum cooking time (OCT), water-absorbing capacity (WA) and cooking loss of fortified pasta samples.

  1. 1 Must discuss well. Need to give clarity on how the sensory quality is affected and why?

A: we added some lines of description to give more clarity

  1. Must explain clearly OCT, WA, Cooking loss

A: we added some lines of description to give more clarity

  1. In section 3.2 give the subsections of 3.2.1 Total phenolic content, 2.3.2. fatty acids composition etc..

A: Paragraph 3.2 “Chemical characterization of raw material and functional pasta” , in results has been divided into 5 subparagraphs , each for the results of the different analyzes

  1. The discussion is a must for section 3.2. The total fatty acids and their health issues should be discussed and mentioned why there is variation among them.                                                                                                     A: There is a comment on the quantities of fatty acids found in pasta, as well as a comparison with data from the literature.

We recall that these were deoiled seed flours, therefore the oil contribution of the flours is very small, as mentioned in the manuscript, it is only 8% of the total dry weight. We know the different activities of omega 3 and omega 6 fatty acids, but this manuscript does not want to go into the merits of the functions of the two different fatty acids.

  1. The Amino acid composition is very important, and must be present as the adding the hemp what is the amino acid increased and decreased? What is the nutritional and health influence?                                                A: As suggested, a wider discussion has been made regarding the increase in essential amino acids in particular lysine and threonine on fortified pasta samples.
  2. Need to explain the phenolic acids and their health benefits as the incorporation of Hemp. Hence rewrite

A: Done in discussion

  1. The discussion on minerals is very poor.

A: We improved the discussion for mineral composition of pasta.

  1. The section 4, the discussion should elaborate much

A: Done

  1. Fine-tune the conclusions

 A: We improved the conclusion section

  1. Check for the English quality                                                                          A: the authors improved English form

  1. Check for the typographical and editorial issues Although this study is interesting but some corrections are needed for publication. Some of mandatory comments given below in order to process it further process.
  2. A: done. Thanks for important suggestions.

Thanks to Reviewer n. 2 for his/her opinion on the subject of the paper and also for his/her valuable comments. The authors rewrote portions of the article and hope the reviewer will support the publication of the manuscript on SI of Foods.

Author Response

The revised version of the manuscript is more comprehensive and certainly meets journal style requirements, according to manuscript format and references. Authors have accepted many of the reviewers; recommendations and addressed to most of their suggestions successfully.
Thanks to Reviewer #3 for their positive comments on the quality improvement of the manuscript.

REV #3
1.             English and manuscript writing should be improved. There are some grammatical mistakes.
A: We improved English form.

2.             The manuscript is fortification of pasta by hemp flour with different particle size. It is data collection and predictable that higher hemp flour content in pasta causes higher antioxidant activity and other properties. It was better to investigate the effect of cooking on the phenolic content of cooked pasta.  How much of phenolic, amino acids and minerals were leached out during cooking?
A: data relating to the loss of micro and macronutrients during cooking have been collected.
We have not presented them in order not to make the article too heavy.
We would insert more the tables relating to the bio-constituents in  raw pasta and in cooking waters, this would have meant increasing the manuscript a lot.
In our opinion, it was more important to define in the manuscript the quality of the product which is then consumed as food.

3.             Why the phenolic content of Hemp 1 was lower than Hemp 2? What was the effect of particle size on their properties?
A: the slight difference in the polyphenolic content between Hemp 1 and Hemp 2 refers exclusively to the bound polyphenols.
Phenolics in insoluble forms are covalently bonded to structural components of the cell wall such as cellulose, hemicellulose, lignin, pectin.
Activities of bound phenols are numerous and for this reason their study is important. In fact, they perform important functions in the protection against the invasion of pathogens, they have antibacterial, antifungal and antioxidant functions, in the body they have functions on the modulation of the intestinal microbiome.

The higher content in the smaller grain size flour could be linked to the better possibility of making these bioactives available, therefore more easily extractable from the matrix.

4.             Please explain the effect of flour particle size on pasta properties. More discussion is needed.
A: The Authors discussed and improved the paragraph. Thank you for the comment. 

5.  Line 144: overall quality as taste and odour (OQS) or Over Quality Score? 
A. The authors have improved the sentence for a better understanding of the text

6.             Line 160: Ciclope or Ciclope flour? 
A: is written……….Ciclope, Hemp 1 and Hemp 2 flours……..
They are all flours

7.             Line 242 and 273: Usually don’t use the number at the beginning of the sentence. 
A: the authors corrected the highlighted errors

8.             Line 270: Antiradical properties or antioxidant? 
A: the DPPH assay measures the antiradical activity

9.             Line 338: “wheat pasta with non-gluten flours may affect the gluten protein network development” which component had an influence on protein function or adhesiveness and fibrous sensation? Please explain and discus the results.  
A: Gluten inevitably affects the adhesiveness and sensation of fibrousness.
The authors changed the text as follows:

……Absence of proteins such as glutenins and gliadins, responsible for the formation of gluten, has inevitably influenced the characteristics of the product. CTRL recorded the greatest overall quality mainly in terms of firmness, adhesiveness and OQS (Table 4); the enrichment of durum wheat pasta with non-gluten flours may affect this parameter, resulting in an increase of adhesiveness and fibrous sensation. Odor and taste in fortified pasta a different substitution levels resulted similar to CTRL pasta, suggesting the particle size of hemp did not affect these traits. The best results in terms of OQS was obtained using Hemp 2 at 7.5% of substitution…..

10.       Please check Line 375: Hemp 1 e Hemp 2.
A. the authors have improved the sentence for a better understanding of the text

Thanks to Reviewer n. 3 for his/her opinion on the subject of the paper and also for his/her valuable comments. The authors rewrote portions of the article and hope the reviewer will support the publication of the manuscript on SI of Foods.

Round 2

Reviewer 3 Report

Title: Hemp flour particle size affects quality and nutritional profile of the enriched functional pasta

Manuscript ID: foods-2176123

 Dear Editor

This manuscript is modified and well written. However, I suggest minor revision before publication according to the comment:

-          The TPC content and antioxidant activity of pasta fortified with Hemp1 was higher than Hemp2. Would you please explain the role of particle size on antioxidant activity? Because the same percentage of Hemp flour was used. Can we conclude that the reduction in particle size of flour increased the release of TPC? Which Hemp particle size do you suggest for fortification of pasta?

Author Response

Title: Hemp flour particle size affects quality and nutritional profile of the enriched functional pasta

Manuscript ID: foods-2176123

Dear Editor

This manuscript is modified and well written. However, I suggest minor revision before publication according to the comment:

  • The TPC content and antioxidant activity of pasta fortified with Hemp1 was higher than Hemp2. Would you please explain the role of particle size on antioxidant activity? Because the same percentage of Hemp flour was used. Can we conclude that the reduction in particle size of flour increased the release of TPC? Which Hemp particle size do you suggest for fortification of pasta?

The manuscript was revised on the basis of the latest suggestions.

Antiradical activity and TPC were higher in Hemp 1, in the flour and in the fortified pasta samples at the different percentages.

Analysis of the phenolic component determined with HPLC-ESI/QTOF-MS both in the free form and in the bound form, are in agreement with the TPC values.

The authors noticed small typing errors in table 8, reported in the correct form.

Data available do not allow us to give an explanation of the different phenolic content in the two flours which differ only in the size of the particles.

As suggested in the discussion and finally in the conclusions, taking into account the sensorial, rheological and nutritional qualities of the fortified pastes, the best replacement percentage is 7.5%.

As shown, the pasta obtained with the replacement of 7.5% of Ciclope semolina with Hemp 2 flour showed a better profile in minerals, amino acids and a higher quantity of polyunsaturated fatty acids; on the contrary, the pasta obtained from the same percentage of substitution but with Hemp 1 flour showed a better phenolic profile, TCP and a better anti-radical activity.

Thanks again
